# Plasma Amino Acids May Improve Prediction Accuracy of Cerebral Vasospasm after Aneurysmal Subarachnoid Haemorrhage

**DOI:** 10.3390/jcm11020380

**Published:** 2022-01-13

**Authors:** Ernest Jan Bobeff, Malgorzata Bukowiecka-Matusiak, Konrad Stawiski, Karol Wiśniewski, Izabela Burzynska-Pedziwiatr, Magdalena Kordzińska, Konrad Kowalski, Przemyslaw Sendys, Michał Piotrowski, Dorota Szczesna, Ludomir Stefańczyk, Lucyna Alicja Wozniak, Dariusz Jan Jaskólski

**Affiliations:** 1Department of Neurosurgery and Neuro-Oncology, Medical University of Lodz, Barlicki University Hospital, Kopcinskiego St. 22, 90-153 Lodz, Poland; karol.wns@gmail.com (K.W.); mmpiotrowskimd@gmail.com (M.P.); dariusz.jaskolski@umed.lodz.pl (D.J.J.); 2Department of Structural Biology, Medical University of Lodz, 90-419 Lodz, Poland; malgorzata.bukowiecka-matusiak@umed.lodz.pl (M.B.-M.); izabela.burzynska-pedziwiatr@umed.lodz.pl (I.B.-P.); dorota.szczesna@umed.lodz.pl (D.S.); lucyna.wozniak@umed.lodz.pl (L.A.W.); 3Department of Biostatistics and Translational Medicine, Medical University of Lodz, Mazowiecka 15 Street, 92-215 Lodz, Poland; konrad.stawiski@gmail.com; 4Department of Radiology, Barlicki Memorial Teaching Hospital, Medical University of Lodz, Kopcinskiego 22 Street, 90-153 Lodz, Poland; magdalena.kordzinska@gmail.com (M.K.); ludomir.stefanczyk@umed.lodz.pl (L.S.); 5Laboratorium Diagnostyczne Masdiag, ul. Żeromskiego 33, 01-882 Warszawa, Poland; konrad.kowalski@masdiag.pl (K.K.); przemyslaw.sendys@masdiag.pl (P.S.)

**Keywords:** aneurysmal subarachnoid haemorrhage, cerebral vasospasm, plasma amino acids, predictors, high performance liquid chromatography-mass spectrometry, Hunt-Hess scale

## Abstract

Aneurysmal subarachnoid haemorrhages (aSAH) account for 5% of strokes and continues to place a great burden on patients and their families. Cerebral vasospasm (CVS) is one of the main causes of death after aSAH, and is usually diagnosed between day 3 and 14 after bleeding. Its pathogenesis remains poorly understood. To verify whether plasma concentration of amino acids have prognostic value in predicting CVS, we analysed data from 35 patients after aSAH (median age 55 years, IQR 39–62; 20 females, 57.1%), and 37 healthy volunteers (median age 50 years, IQR 38–56; 19 females, 51.4%). Fasting peripheral blood samples were collected on postoperative day one and seven. High performance liquid chromatography-mass spectrometry (HPLC-MS) analysis was performed. The results showed that plasma from patients after aSAH featured a distinctive amino acids concentration which was presented in both principal component analysis and direct comparison. No significant differences were noted between postoperative day one and seven. A total of 18 patients from the study group (51.4%) developed CVS. Hydroxyproline (AUC = 0.7042, 95%CI 0.5259–0.8826, *p* = 0.0248) and phenylalanine (AUC = 0.6944, 95%CI 0.5119–0.877, *p* = 0.0368) presented significant CVS prediction potential. Combining the Hunt-Hess Scale and plasma levels of hydroxyproline and phenylalanine provided the model with the best predictive performance and the lowest leave-one-out cross-validation of performance error. Our results suggest that plasma amino acids may improve sensitivity and specificity of Hunt-Hess scale in predicting CVS.

## 1. Introduction

Aneurysmal subarachnoid haemorrhages (aSAH) account for 5% of strokes, and characteristically it affects younger people. Unlike other strokes, its case fatality varies between low to middle- and high-income countries [1], which possibly reflects differences in medical management substantially contributing to the patients’ outcome [2]. Although the crude global incidence has declined in recent decades [3], aSAH continues to place a great burden on both patients and their families.

Cerebral vasospasm (CVS) is one of the main causes of death after aSAH. It is responsible for delayed cerebral ischemia (DCI), an umbrella term which encompasses: symptomatic CVS, delayed ischemic neurologic deficit (DIND), and asymptomatic delayed cerebral infarction. CVS is usually diagnosed between day 3 and 14 after bleeding and is characterised by long-term changes in morphology of arterial walls. The pathogenesis remains poorly understood, despite numerous hypotheses concerning the role of blood components, endothelial dysfunction, and vascular innervation, potentially leading to deleterious contraction of cerebral arterial smooth muscles [4].

Factors influencing prognosis after aSAH have been a long-standing issue that has gained special attention since development of microsurgical techniques [5]. The main role of surgical and endovascular interventions remains prevention from rebleeding, thus, its benefits must be carefully weighed against risks. This notwithstanding, it also facilitates CVS prophylaxis and conservative treatment, including vasodilation and haemodynamic augmentation. In 1968 Hunt and Hess stressed that mortality of patients after aSAH is particularly associated with their condition at the time of surgery [6]. Importantly, they introduced a risk stratification Hunt and Hess (HH) scale for selecting the optimal time for surgical intervention, used to date. According to the recent meta-analysis, patients with HH grade I have lower risk of developing CVS [7]. In poor-grade aSAH patients urgent surgical clipping provided little benefit considering potential intraoperative injury (i.e., aneurysm rupture, difficult exposure, prolonged temporary clipping, blood loss and transfusions, and intraoperative hyper/hypotension) [8]. The new recommendation endorses endovascular coiling in the elderly, poor-grade aSAH, and aneurysms of the basilar tip [9]. Still, it is recommended that treatment decisions be made together with the patient, after receiving recommendations from the multidisciplinary team [10].

Diagnosis of CVS after aSAH requires presence of both, persisting neurologic deficit with delayed onset and relevant changes in radiological imaging. The accepted ancillary tests include digital subtraction angiography (DSA) and transcranial Doppler ultrasound (TCD), whereas computed tomography angiography (CTA) was claimed to overestimate the degree of stenosis [11]. TCD provides rapid, non-invasive, real-time measures of cerebrovascular function, and therefore it is currently considered a standard of care.

Neuromonitoring is a developing interdisciplinary field that helps in early detection of patients at risk of symptomatic CVS, which is critical to successful treatment. Continuous monitoring of brain function requires invasive procedures aimed at maximization of cerebral blood flow (CBF) [12]. On the other hand, there are some novel and less invasive strategies being studied [13]. By way of illustration, in a recent study we proposed a single non-invasive urine biomarker for early diagnosis and monitoring of DCI [14].

Prevention and treatment of CVS remains a matter of debate. Established approaches are the administration of nimodipine initially intravenously then orally and maintenance of euvolemia, as well as treatment with hemodynamic augmentation therapy or endovascular therapy with vasodilators and angioplasty balloons [8]. A recent study also suggested the potential use of heparin [15].

The main objective of this study was to verify whether concentration of plasma amino acids demonstrate diagnostic and prognostic value in predicting CVS in patients after aSAH. Small molecule metabolites fulfil vital functions throughout the body, and thus constitute a favourable target for biomarker research in various diseases. Some authors suggested that haemorrhagic cerebrovascular disease may induce hypermetabolic state of the organism [16,17]. In particular, surgery for SAH showed to provoke a catabolic response and defective utilization of exogenous nitrogen [18], what raised our suspicion that it might impact plasma levels of amino acids.

## 2. Materials and Methods

We performed a prospective observational case-control study in patients after aSAH treated at the Department of Neurosurgery, Barlicki University Hospital in 2016 and 2017. The study was carried out in compliance with the Helsinki Declaration and ethics approval was acquired from the Ethics Committee of the Medical University of Lodz.

### 2.1. Participants

The inclusion criteria were as follows: 18 to 70 years of age; presence of aneurysm confirmed on computed tomography angiography (CTA) or digital subtraction angiography (DSA); aSAH grade I-III according to the Hunt-Hess scale; preoperative chest X-ray, resting electrocardiogram and routine laboratory tests within normal limits; written informed consent obtained from the patient. The study group consisted of 35 patients after aSAH, out of 122 cases treated at our institution during the enrolment period. All were managed according to the current standard-of-care guidelines [9] and in each case aneurysm repair was performed within 48 h of bleeding. In the control group there were 37 age- and sex-matched healthy volunteers. The exclusion criteria for both the study and the control groups were specified as follows: serious systemic disease, malnutrition, altered state of consciousness, presence of malignancy, multiple aneurysms, hydrocephalus with the need for external ventricular drainage, and/or prolonged respiratory support (>24 h). Treatment outcome was assessed at 3 and 12 months after SAH and reported using the Glasgow Outcome Scale, GOS. 

### 2.2. Cerebral Vasospasm 

Cerebral vasospasm (CVS) was diagnosed only in case of delayed neurological deterioration (i.e., confusion, decreased level of consciousness, or focal neurologic deficit lasting for at least 1 h) after excluding other causes. CVS was assessed by TCD, and the threshold was blood flow velocity exceeding 120 cm/s in the middle cerebral artery (MCA), and the Lindegaard ratio (MCA/ICA flow velocity) greater than 3. To confirm CVS we performed DSA. All measurements were obtained by an experienced neuroradiologist (LS).

### 2.3. Sample Collection

Fasting peripheral blood samples were collected into commercially available ethylenediaminetetraacetic acid (EDTA) treated tubes on postoperative day one and seven as part of a routine examination. Cells were removed from plasma by centrifugation for 10 min at 7000× *g*. Samples were stored in liquid nitrogen afterwards. The overall duration of this phase was limited to 30 min to minimize its impact on plasma metabolic profile [19]. Determination of metabolites in the collected material was carried out at the Department of Structural Biology, Medical University of Lodz.

### 2.4. Chemicals and Reagents

The methanol used in the procedure was from Sigma Aldrich (Sigma Aldrich Chemie GmbH, Steinheim, Germany). Solution of internal, deuterated standards for amino acids concentration calculation, were produced by CIL (Cambridge Isotope Laboratories, Andover, Tewksbury, MA, USA). For derivatization by butylation of the carboxyl group of the analyte and formation of the butyl ester, 3N HCl in n-butanol (3N Hydrochloric Acid in 1-Butanol) was used from REGIS TECHNOLOGIES, INC.(Austin Avenue, Morton Grove, IL, USA). Formic acid (FA), acetonitrile (ACN), heptafluorobutyric acid (HFBA) ordered from J. T. Baker (Avantor Performance Materials B.V., Deventer, The Netherlands).

### 2.5. Sample Preparation

10 µL of patient plasma with 10 µL internal standard solution were placed in polypropylene deep-well plate. Precipitation and extraction were performed by adding 780 µL of 1 N HCl in methanol. Internal standard solution contains 125 nmol/mL Gly labeled and 25 nmol/mL labeled Ala, Asp, Glu, Leu, Met, Phe, Tyr, Val, Orn, Cit. Samples were mixed on an orbital shaker at 600 rpm, for 10 min, at RT. Precipitation was additionally improved by incubation at −20 °C for 2 h. Samples were centrifuged at 1000 rpm for 5 min. 50 µL of supernatant was transferred to a new 96 well plate with (250 µL total well volume). The contents of the well plate were evaporated to dryness for about 15 min. 25 µL 3N HCl in n-butanol was added to each well. The samples prepared in this way were incubated at 60 °C for 25 min. Evaporated again to dry for about 15 min. Evaporation residue was dissolved in 200 µL H_2_O: MeOH (80:20) + 0.1% FA. The dissolved samples were mixed on a shaker at 600 rpm for 10 min, at RT.

### 2.6. HPLC-MS Analysis

HPLC-MS/MS analyses were performed by ExionLC (AB Sciex) liquid chromatograph equipped with an Exigent autosampler (AB Sciex) coupled with tandem mass spectrometer (4500 QTRAP, AB Sciex) with electrospray ion source (Turbo V, AB Sciex). 2 µL of the dissolved sample was injected for analysis into ACE Excel C18 column with dimensions of 2.1 mm × 50 mm × 1.7 µm. Flow ratio of the column was 0.4 mL/min with the temperature 40 °C. H_2_O with 0.5 mM HFBA and MeOH in proportion with (1:1) ACN constituted respectively, eluents A and B. The time and used gradient of the eluents are shown in Table 1.

### 2.7. Statistics

Statistical analyses were performed using the Statistica software (Version 13; StatSoft, Inc., Tulsa, OK, USA) and R programming language [20]. Due to the lack of normally distributed data (assessed using the Shapiro-Wilk test), we compared continuous variables between two groups with Mann-Whitney’s U test. Wilcoxon signed rank exact test for comparison of paired data. The frequencies of categorical variables were compared using Pearson’s chi-square test, chi-square test with Yates’ correction or two-tailed Fisher’s exact test, when appropriate. For all tests, *p* < 0.05 was deemed to be significant. Due to the risk of overfitting associated with an event-to-variable ratio of less than 10, the multivariate analysis was omitted. The principal component analysis was performed to check for outliers and batch effect.

In predictive model development for CVS, the features were at first preselected based on AUC ROC (area under the receiver operating characteristic) curve. A logistic regression models were developed for significant amino acids, Hunt-Hess score (as reference) and for joint model. The cut-off for the prediction estimate was selected based on maximum value of Youden index. Models were assessed based on their accuracy, sensitivity, specificity, PPV (positive predictive value), NPV (negative predictive value). Lastly, due to lack of external validation group, the overfitting of models was assessed in leave-one-out cross-validation (LOOCV).

## 3. Results

We enrolled 35 patients (20 females, 57.1%) after aSAH who fulfilled the eligibility criteria. All aneurysms were successfully secured within 48 h after initial bleed, by surgical clipping (*n* = 34, 97.1%) or endovascular coiling (*n* = 1, 2.9%; basilar artery aneurysm). The median age was 55 years (IQR 39–62) in the study group and 50 years (IQR 38–56) in the control group. The median Hunt-Hess score at presentation was 3 (IQR 2–3). Good outcome (GOS 4–5) was noted in 18 patients (51.4%) after 3 months and in 23 patients (65.7%) after 12 months. Cerebral vasospasm (CVS) was observed in 18 patients (51.4%), and it was usually diagnosed on postoperative day 5 (IQR 4–6). Symptoms of CVS were as follows: disorientation 17 (94.4%), new or increasing H/A 11 (61.1%), focal neurological deficit 9 (50%), and lethargy 5 (27.8%). More details about the study group clinical features and measurements are provided in Table 2 and Table 3.

All evaluated scales (i.e., Hunt-Hess scale, Fisher scale, World Federation of Neurosurgical Societies scale (WFNS) and Glasgow coma scale (GCS)), differed between patients with and without CVS (all *p* < 0.01). However, only Fisher Scale and GCS differed between patients with poor and good outcome (both *p* < 0.05). Patients with poor outcome had significantly higher RDW-CV on admission (13.9% vs. 12.8%, *p* = 0.016). Characteristically, patients with CVS were hospitalized for twice as long (*p* < 0.001). 

Differential analysis has shown that patients after aSAH, in comparison to healthy controls, featured distinctive amino acids concentration, which was not only shown in the principal component analysis (Figure 1A), but also after the direct comparisons (Figure 1B, Table 4). Interestingly, the concentration of 8 out of 19 studied amino acids was found to be significantly lower in SAH patients, and this remained true after the multiple comparisons’ adjustment. No significant differences were noted between postoperative days one and day seven in the SAH patients (Figure 1C,D), indicating a lack of plasma amino acids concentration change during the first seven days after SAH.

Standardized concentration values measured were shown as heatmap in Figure 2. Correlation matrix of concentration was visualized in Figure 3.

In the patients who developed CVS, hydroxyproline (Pro-OH) concentration was significantly lower on postoperative day one (Figure 4B). Arginine and methionine concentration was higher in those patients on postoperative day seven, but all those differences became insignificant after the multiple comparison adjustment. Generally, no significant differences in the amino acids concentration were noted between patients with and without CVS (Figure 4A,B) as well as good and poor GOS result (Figure 4C,D). However, hydroxyproline (Pro-OH, AUC = 0.7042, 95%CI 0.5259–0.8826, *p* = 0.0248) and phenylalanine (Phe, AUC = 0.6944, 95%CI 0.5119–0.877, *p* = 0.0368) presented significant CVS prediction potential, as shown in Figure 5. 

Based on the analysis of area under the ROC curves for selected amino acids and for prediction of CVS (Figure 5) Phe and Pro-OH turned out to be the only amino acids with significant predictive potential. Since the Hunt-Hess scale is considered one of the reference tools for prediction of CVS after SAH, we compared models based on amino acids and Hunt-Hess scale, which performed roughly similar. Combining both the Hunt-Hess scale and amino acids concentration provided the model with the best predictive performance and the lowest leave-one-out cross-validation of performance error, and thus it seemed to be the most resilient model to overfitting (Table 5, Figure 6B). The final logistic regression model presented with excellent accuracy, sensitivity, and specificity, and therefore allowed us to develop a nomogram for prediction of CVS based on the Hunt-Hess scale and plasma concentration of Phe and Pro-OH (Figure 6A). The nomogram can be easily applied in clinical practice and serves for risk stratification of patients beyond established clinical risk markers.

## 4. Discussion

Early identification of patients at risk for CVS who could benefit from aggressive intervention is extremely important to improve neurological outcomes. We intended to find a change in the amino acids’ concentration in the blood before clinically overt CVS. Hence, the test was conducted just after aneurysm repair and then at the point at which CVS is likely to occur. Our results support the hypothesis that plasma amino acids present diagnostic and predictive value in patients after aSAH. Despite advances in the diagnosis and treatment of aSAH, effective therapeutic interventions are still limited. CVS is one of the major problems responsible for the high mortality and morbidity associated with SAH [21]. It probably accounts for the impaired cerebral metabolism that has been shown in various studies [22,23,24]. We reasoned that the metabolites altered in this process could be used as a clinical diagnostic tool. Based on the area under the ROC curve analysis, the concentration of hydroxyproline and phenylalanine in the blood on day one after the operation were considered adequate for CVS prognostication, and were further used to increase accuracy of Hunt-Hess scale. The amino acid-based model for CVS prediction in aSAH was comparable to the commonly employed reference tool (i.e., the Hunt-Hess scale). The presented combined model showed the best performance in predicting CVS, and the nomogram provided visualized risk prediction (Table 5, Figure 5 and Figure 6).

What was observed is that both aSAH and surgery induce a hypermetabolic state of the body. One proposed mechanism for this phenomenon is dysregulation of the vegetative system with increased sympathetic activity [16]. Tuoho et al. (1990) found increased resting energy expenditure in patients with haemorrhagic cerebrovascular disease at the acute stage. They also showed that the rate of protein consumption was higher in patients after SAH grade three to four according to Fisher compared to patients in grade one to two. Similarly, Suojaranta-Ylinen et al. (1996) pointed to the effect of adrenergic stimuli on the central nervous system as an underlying factor, and demonstrated that increased amino acid infusion had no effect on amino acid exchange (suggesting that other factors are also involved) [17].

Amino acids are small-molecule metabolites and play an important role in neuronal circuitry, development, maintenance, growth, and survival of neurons in the brain. Molecular movement across the blood-brain barrier is mediated via diffusion, active transport, and carrier-mediated transport, which is particularly used by amino acids. Recently, we demonstrated differences in the levels of certain amino acids in the blood of patients with glioblastoma compared to healthy subjects [25]. The current study, comparing SAH patients with healthy controls, showed that the plasma levels of alanine, glutamic acid, glycine, leucine, methionine, proline, hydroxyproline, tyrosine, and valine differed significantly. Disrupted uptake via blood brain barrier was observed in various brain pathologies including tumours and aSAH [26,27], and it may constitute a common mechanism for the above-mentioned observations. It is interesting to note that there was no significant difference between amino acids concentration on day one and seven post-operation in the SAH group.

Hydroxyproline is a common non-proteinogenic amino acid found predominantly in collagen and elastin–major arterial wall proteins [28]. Unlike other amino acids, it shows smaller biological variation in CSF than in serum [29]. Post-translational modification of hydroxyproline is catalysed by the prolyl subunit of 4-hydroxylase alpha-3 (P4HA3), a key enzyme in collagen synthesis [30]. Increased accumulation of intracellular hydroxyproline and reduced procollagen synthesis in skin fibroblasts were found in patients harbouring cerebral aneurysms [31]. It seems to be consistent with other observations that hydroxyproline and proline are critical for the mechanical strength of collagen [32]. The results of our study demonstrated decreased plasma level of hydroxyproline in patients after aSAH on the postoperative day one compared to controls. In patients who developed CVS later, the level of hydroxyproline was even lower. Similarly, Sokol et al. (2017) reported that the CSF level of hydroxyproline did not increase after aSAH, unlike the rest of amino acids in their group [33]. In an attempt to explain this phenomenon, the authors concluded that the lack of increase in CSF after aSAH might be related to the high binding of hydroxyproline to proteins in human erythrocytes [34]. Presumably, hydroxyproline deficiency can reduce mechanical strength of arterial walls and thus predispose to aneurysm formation and an increased vulnerability to vasospasm. It is possible that in patients with a genetic predisposition, blood levels of hydroxyproline may be a predictor of aSAH as well. Interestingly, the theory that plasma amino acids concentration may be involved in etiopathogenesis of SAH has been already formulated [35,36], although the authors focused on the level of homocysteine, which we did not measure.

Phenylalanine is an essential amino acid involved in protein biosynthesis. It contains a distinctive aromatic ring which makes it one of the biggest proteinogenic amino acid. It is also converted to tyrosine, a precursor of monoamine neurotransmitters. Both phenylalanine and tyrosine are substrates for tyrosine hydroxylase, the enzyme catalysing the rate-limiting step in catecholamine synthesis. The first conversion catalysed by phenylalanine hydroxylase is the source of L-tyrosine, but is also the first step in tyrosine/phenylalanine catabolism. Tyrosine can be converted into 3,4-dihydroxy-L-phenylalanine (L-DOPA), which, in the next step, is converted into dopamine, norepinephrine, and epinephrine [37]. Tyrosine is the preferred substrate; consequently, unless its concentration is abnormally low, changes in phenylalanine concentration does not affect catecholamine synthesis. Unlike tyrosine, phenylalanine does not exhibit substrate inhibition, therefore, its high concentration does not inhibit catecholamine synthesis [37]. Clinical and experimental studies have shown that patients after SAH exhibited significant activation of the sympathetic nervous system [38]. Minegishi et al. (1987) demonstrated that plasma norepinephrine metabolite levels can serve as a prognostic discriminator for patients after aSAH [39]. Catecholamine surge causes activation of the sympathetic nervous system and may induce arrhythmias, neurogenic pulmonary edema, injury to the hypothalamus and brainstem, and is probably associated with the development of CVS [40]. We observed significantly higher levels of phenylalanine on day one post-operation in patients who later developed CVS (*p* = 0.049), what is consistent with the literature cited above. 

An important contribution to the understanding of amino acid metabolism in brain pathology was based on studies concerning composition of interstitial fluid (ISF) and cerebrospinal fluid (CSF). Zetterling et al. (2009) reported increased concentration of eight amino acids (alanine, asparagine, glutamine, isoleucine, leucine, phenylalanine, serine, and tyrosine) in ISF after SAH [41]. Li et al. (2019) found that increased amino acid levels in CSF after SAH was associated with unfavorable outcome [42]. Particularly, high levels of asymmetric and symmetric dimethylarginines (ADMA, SDMA) in CSF were associated with poor outcome [43]. The same authors confirmed that plasma L-arginine/ADMA ratio is negatively associated with outcome, which was earlier suggested by Staalsø et al. [44]. Although we did not measure the ratio, our results showed that the absolute value of plasma arginine was higher in patients with poor outcome and those who developed CVS. Von Holst and Hagenfeldt (1985) demonstrated increased amino acids concentration in CSF after SAH and proposed mechanisms leading to this phenomenon [45]. The initial increase may be due to extravasation of blood into the CSF and therefore depends on the amount of bleeding. It also might be due to the increased amino acid turnover in response to injury and proteolysis, which initiates a repair process that is in turn accompanied by increased protein synthesis [46]. The widespread activation often extends beyond the area of actual brain damage [47]. When combined with impaired transport across the blood-brain barrier it results in increased amino acids concentration in CSF and ISF and decreased amino acids concentration in plasma. Interestingly, we also observed decreased levels of amino acids in plasma. Other biological compounds have been studied as well. Bellapart et al. (2014) found that endothelin-1 level is higher in plasma than in CSF on the day five after SAH in patients who developed CVS [48]. On the other hand, CVS occurs on average at that day, whereas the idea of predictive biomarker for CVS is to indicate patients at risk beforehand. In addition, the plasma biomarker would be more desirable since not all of the patients after aSAH require CSF study.

Despite the fact that our results have empirically demonstrated the possible utility of plasma amino acid analysis in patients after aSAH, there are some limitations of this study that can be addressed in developing a new model. First, the proposed measurements are not part of the basic tests performed in a hospital, and thus the results cannot yet be implemented in clinical practice. However, they may provide guidance for subsequent clinical studies. Given the magnitude of disease burden, our sample size remains somewhat inadequate to expect similar results in a population-based study. Predictive models built on small groups may be particularly susceptible to overfitting. One reason for clinicians not using them is the lack of external validation so that they do not know if it predicts the risk precisely in other patients. The main sources of potential bias can be dietary habits, stimulant use, and often under-recognized common chronic diseases. 

## 5. Conclusions

The purpose of the current study was to determine whether plasma amino acids may help to differentiate between patients with and without increased risk of developing CVS after aSAH. The results presented here can potentially be used in a clinical setting along with reference tools. One of the more significant findings to emerge from this study is that plasma levels of hydroxyproline and phenylalanine may improve sensitivity and specificity of Hunt-Hess scale in predicting CVS. It is suggested that the association of these factors should be further investigated.

## Figures and Tables

**Figure 1 jcm-11-00380-f001:**
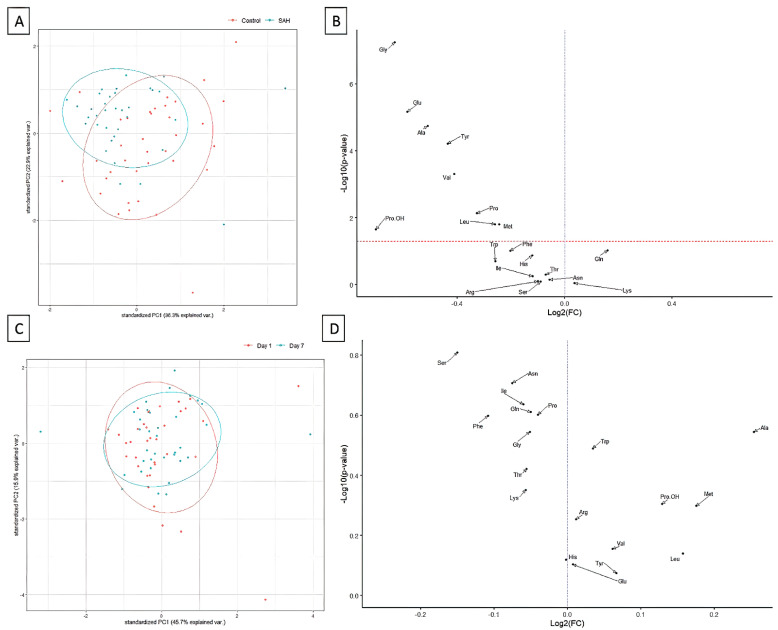
Principal component analysis and differential concentration analysis (volcano plots) showing the differences in amino acids concentration between SAH and control patients (**A**,**B**) and between day one and day seven post-operation (**C**,**D**). The relevant *p*-values are available in Table 4.

**Figure 2 jcm-11-00380-f002:**
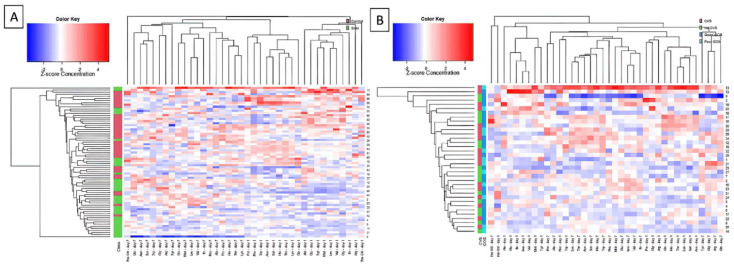
Heatmaps presenting standardized concentration of selected amino acids with hierarchical clustering across both amino acids and samples. (**A**) presents the comparison between SAH and control patients. SAH patients seem to cluster together. (**B**) presents intragroup clustering of patients with distinctive GOS and with or without CVS.

**Figure 3 jcm-11-00380-f003:**
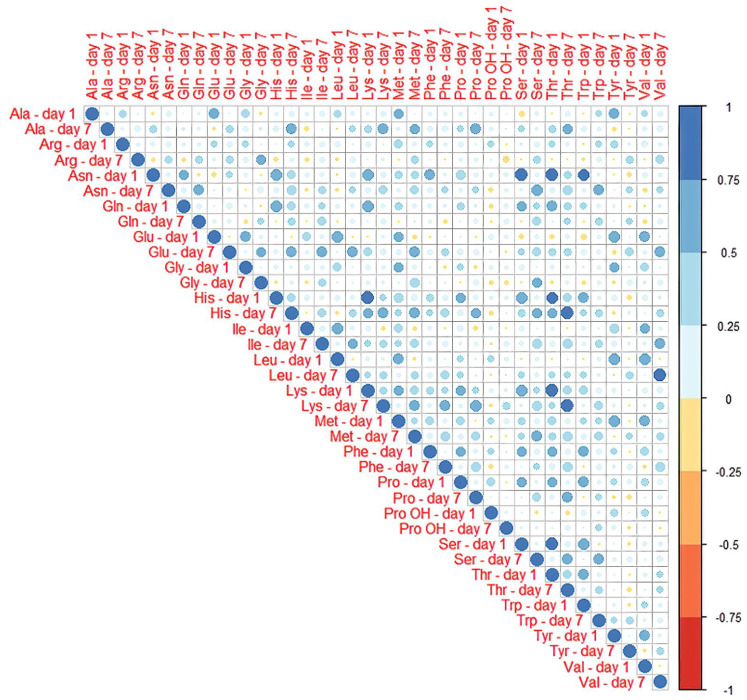
Correlations between amino acids concentration in patients with SAH and between postoperative day one and day seven. Scale presents the value of R coefficient, indicating the strength of the correlation.

**Figure 4 jcm-11-00380-f004:**
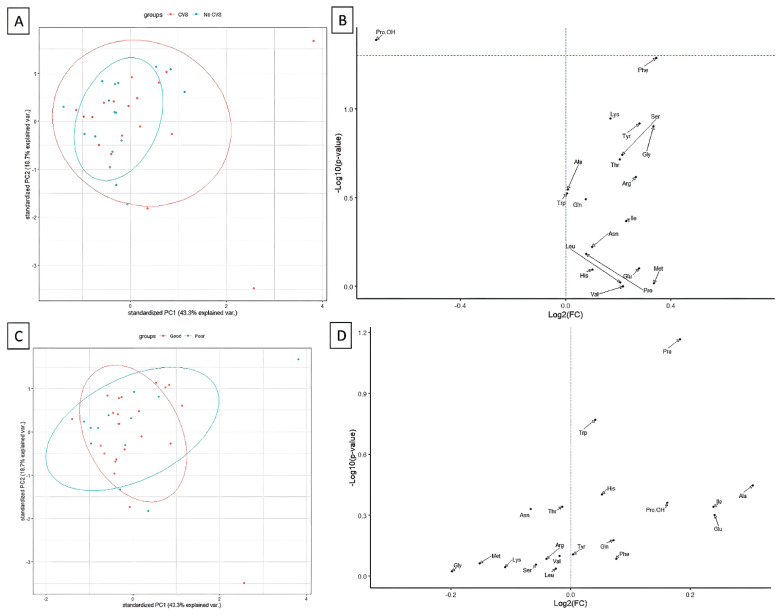
Principal component analysis and differential concentration analysis (volcano plots) showing the differences in amino acids concentration on postoperative day 1 between patients who developed CVS or not (**A**,**B**) and good or bad GOS (**C**,**D**). The relevant *p*-values are available in Table 4.

**Figure 5 jcm-11-00380-f005:**
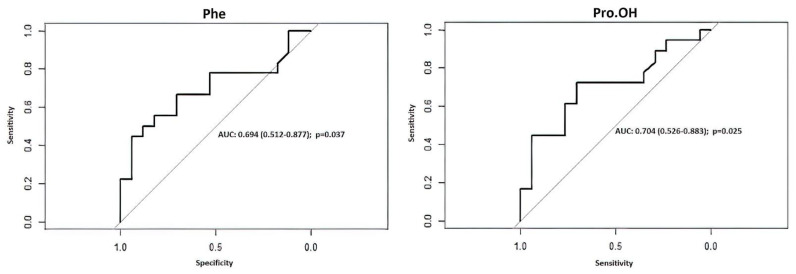
ROC analysis of selected amino acids in prediction of CVS has proved the predictive potential of phenylalanine (Phe) and hydroxyproline (Pro-OH). Area under the curves (AUC) were provided with their 95% confidence interval in parenthesis.

**Figure 6 jcm-11-00380-f006:**
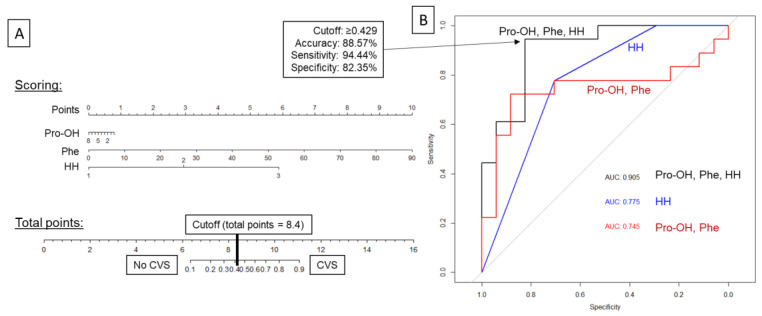
ROC curves of developed logistic regression models with the nomogram for the best model. As shown in (**B**), the combined model outperformed the model based on amino acids concentration alone (Pro-OH, Phe) and the model based on Hunt-Hess (HH) scale. The best cutoff point for the logistic regression estimate was calculated based on Youden index and marked with arrows and performance characteristics. (**A**) allows the utilization of the combined model. After scoring individual predictors, if the sum exceeds 8.4, one can predict CVS with a sensitivity of 94.44% and specificity 82.35%.

**Table 1 jcm-11-00380-t001:** Time and percentages of used eluents during HPLC-MS analysis.

Time of Elution	Eluent A	Eluent B
(min)	(%)	(%)
0	80	20
7	50	50
7.1	5	95
9	5	95
9.1	80	20
11	80	20

**Table 2 jcm-11-00380-t002:** Study group characteristics.

Characteristic	*n* = 35
Male	15 (42.9%)
Age	55 (39–62)
Length of stay at the dept. of neurosurgery	15 (10–21)
Aneurysm location	
Anterior communicating artery	19 (54.3%)
Middle cerebral artery	9 (25.7%)
Internal carotid artery	2 (5.7%)
Posterior communicating artery	2 (5.7%)
Basilar artery	1 (2.9%)
Posterior inferior cerebellar artery	1 (2.9%)
Pericallosal artery	1 (2.9%)
Clinical state on admission	
Hunt-Hess score	3 (2–3)
1	5 (14.3%)
2	11 (31.4%)
3	19 (54.3%)
Fisher score	4 (2–4)
WFNS score	2 (1–4)
GCS score	14 (10–15)
Nuchal rigidity	29 (82.9%)
Neurological deficit	3 (8.6%)
Surgical clipping	34 (97.1%)
Endovascular coiling	1 (2.9%)—BAA
Cerebral vasospasm CVS	18 (51.4%)
Disorientation	17 (94.4%)
New or increasing headache	11 (61.1%)
Focal neurological signs	9 (50%)
Lethargy	5 (27.8%)
Occurrence of CVS (postoperative day)	5 (4–6)
Treatment outcome	
GOS at discharge	4 (3–4)
GOS at discharge ≥4 (good outcome)	18 (51.4%)
GOS at one year	4 (2–4)
GOS at one year ≥4 (good outcome)	23 (65.7%)
mRS	2 (2–5)

BAA, basilar artery aneurysm; CVS, vertebral vasospasm; GOS, Glasgow outcome scale; mRS, modified Rankin scale; WFNS, World Federation of Neurosurgical Societies.

**Table 3 jcm-11-00380-t003:** Intergroup differences analysis of data of 35 patients after aneurysmal subarachnoid hemorrhage. Blood samples were collected on post-operative day 1 and 7. All concentrations are given in nmol/mL, and were averaged based on two measurements. * For all tests, *p* < 0.05 was deemed to be significant.

	Poor Outcome after 12 Months GOS 1–3, *n* = 12	Good Outcome after 12 Months GOS 4–5, *n* = 23	*p* Value	CVS *n* = 18	No CVS *n* = 17	*p* Value	Control Group
Age	60.5 (36.25–62.75)	52 (39–61)	0.424	56 (46–63.5)	55 (33.5–61)	0.215	50 (38–56)
Male	7 (58.3%)	8 (34.8%)	0.282	10 (55.6%)	5 (29.4%)	0.176	18 (48.6%)
LOS	18.5 (14.5–42.5)	12 (9–19)	0.033 *	20.5 (15–30.25)	10 (8.5–14)	<0.001 *	
Hunt-Hess score	3 (2.25–3)	2 (2–3)	0.140	3 (2.75–3)	2 (1–3)	0.002 *	
Fisher score	4 (4–4)	3 (2–4)	0.009 *	4 (3–4)	3 (1.5–4)	0.004 *	
WFNS score	4 (2–4)	1 (1–4)	0.051	4 (2–4)	1 (1–2)	0.003 *	
GCS score	11 (9–13.8)	15 (12–15)	0.026 *	11.5 (9–14)	15 (14–15)	0.004 *	
Nuchal rigidity	11 (91.7%)	18 (78.3%)	0.640	17 (94.4%)	12 (70.6%)	0.088	
Neurological deficit	3 (25%)	0	0.034 *	2 (11.1%)	1 (5.9%)	1.000	
POD-1 Alanine	153.3 (103.7–186.4)	156.8 (120–233.4)	0.362	151 (105.3–192.3)	162.2 (126.1–226.3)	0.287	266.6 (206.3–311.8)
POD-7 Alanine	186.3 (109.4–249.5)	140.6 (109.7–163.5)	0.132	120.9 (103.9–218.4)	147.4 (113.7–175.5)	0.636	
POD-1 Arginine	39.9 (35.1–45.3)	37.1 (32–50.5)	0.824	41.4 (34.6–50.1)	36.8 (30.5–47.6)	0.245	39.6 (32.8–57.3)
POD-7 Arginine	48.1 (36.3–53.8)	39.1 (30.8–48.6)	0.151	47.8 (39–51.9)	36.9 (28.2–46.4)	0.038 *	
POD-1 Asparagine	17.6 (15.7–22.6)	19.2 (16.7–22)	0.461	18.8 (16.5–24.3)	18.5 (15.8–21.3)	0.590	19.2 (13.6–26.4)
POD-7 Asparagine	22.7 (17.1–25.9)	19.8 (16.6–21.6)	0.420	20.8 (16.8–25.6)	19.6 (16.6–23.8)	0.708	
POD-1 Glutamine	938.3 (823.9–1168.5)	960.9 (867.1–1340.5)	0.668	1053.7 (829.7–1345.3)	924.5 (875.2–1107.9)	0.318	915.8 (774.1–1071.1)
POD-7 Glutamine	1163.9 (859.5–1301.4)	1094 (963.2–1238.6)	0.771	1048.8 (935.7–1301.4)	1161.1 (952.9–1247.1)	0.757	
POD-1 Glutamic acid	41.7 (38.8–56.6)	50.7 (39.2–62.7)	0.503	44 (38.3–60.9)	42.1 (39.7–61.1)	0.782	71.2 (61.2–92.5)
POD-7 Glutamic acid	51.2 (38.7–61.9)	47.8 (37.9–58.6)	0.572	53 (41.9–65.9)	46.5 (34.9–55.8)	0.103	
POD-1 Glycine	102.1 (86.4–134.6)	107.6 (91.6–131.4)	0.932	114.9 (98.4–134.4)	93.9 (83–125.5)	0.126	178.8 (156.8–215.6)
POD-7 Glycine	140.2 (90.7–196.3)	111.9 (89.1–146.4)	0.362	140.2 (99.9–171.8)	100.1 (83.4–144.8)	0.057	
POD-1 Isoleucine	28.6 (21–35.5)	34.6 (22.8–44.5)	0.461	33.9 (23.7–44.7)	31 (21.5–39.5)	0.424	36 (23.7–44.7)
POD-7 Isoleucine	30.4 (23.9–47.6)	38.2 (28.7–48.2)	0.440	37.2 (26.3–52.9)	35.2 (27.6–44.5)	0.568	
POD-1 Leucine	96.4 (79.2–157.8)	105.6 (78.3–124.2)	0.905	101.5 (83.5–130.1)	105.6 (71.3–132)	0.935	138.1 (102.2–175.3)
POD-7 Leucine	97.1 (88.8–126.9)	94.6 (80.8–126.7)	0.824	98.1 (88.6–135.7)	94.6 (76.5–120.3)	0.369	
POD-1 Lysine	69.7 (59–83.3)	69.3 (61.6–78.2)	0.905	74.6 (64.5–82.5)	64.4 (55.4–74.1)	0.110	75.4 (56.7–87)
POD-7 Lysine	80.5 (53.6–90)	72.8 (57.2–86.7)	0.771	76.3 (61–87.8)	78.6 (57.2–92.3)	0.807	
POD-1 Methionine	12.5 (10.1–18.8)	11.9 (9.1–18.2)	0.851	12.5 (9.5–19.2)	11.9 (9.7–18.1)	0.961	17 (13.7–20.2)
POD-7 Methionine	12.8 (11–17.6)	11.2 (9.1–13.9)	0.073	12.8 (10.6–17.3)	11.2 (8.9–13)	0.032 *	
POD-1 Phenylalanine	37.6 (24.3–45.3)	39.2 (31–42.6)	0.824	42 (32.3–53.6)	33.5 (28.5–40.1)	0.049 *	51.2 (8.4–65.3)
POD-7 Phenylalanine	44.2 (22.7–63.4)	40.9 (34–49.7)	0.503	46.1 (32.7–55.9)	39.7 (33.6–44.9)	0.184	
POD-1 Proline	47.2 (39.6–76.3)	62.2 (53.6–91)	0.068	64.5 (45.4–91.5)	58.5 (48.2–66.4)	0.660	85.6 (63.6–101.6)
POD-7 Proline	67.7 (50.3–125.1)	71.6 (54.5–85.5)	0.878	69.4 (51.2–81.3)	71.6 (53–89.4)	0.708	
POD-1 Hydroxyproline	2.7 (2–3.9)	2.9 (2–4.3)	0.440	2.1 (1.9–4.2)	3.2 (2.4–5.1)	0.038 *	4.5 (2.4–7.3)
POD-7 Hydroxyproline	2.6 (0.8–3.4)	3.2 (0.2–4.3)	0.440	2.2 (0.2–3.3)	3.6 (2.3–4.5)	0.053	
POD-1 Serine	39.3 (31.5–53.9)	42.2 (34.5–51.4)	0.878	42.3 (36.8–53.4)	35.8 (30.4–51)	0.184	46.2 (31.9–57.5)
POD-7 Serine	47.8 (34.1–58.8)	43.9 (39.5–52.2)	0.986	48.3 (40.4–59.3)	43.7 (36.6–52.1)	0.335	
POD-1 Threonine	32 (22.8–39.6)	34.7 (25.7–40.7)	0.461	35.1 (29.4–41.2)	27.8 (24.9–39.5)	0.195	35.7 (25.8–48.6)
POD-7 Threonine	33.8 (26.5–45.7)	33.6 (28.2–43)	0.878	36.7 (28.9–46.2)	31.9 (26.7–42.7)	0.386	
POD-1 Tryptophan	23.6 (21.6–26.1)	26.5 (23.7–31.2)	0.172	23.7 (21.2–31.4)	26.5 (25.7–29.6)	0.303	30.7 (19.9–37.8)
POD-7 Tryptophan	22.9 (18.3–31.8)	23.9 (18.2–33.1)	0.986	24 (18.5–31.1)	22.8 (18.2–36.1)	0.757	
POD-1 Tyrosine	31.3 (25.3–40.6)	29.5 (24.8–36.6)	0.771	32.9 (26.1–41.5)	27.3 (24.6–32.5)	0.118	42 (35.8–52.4)
POD-7 Tyrosine	33.9 (27.4–37.8)	29.5 (24.2–37.8)	0.420	33.6 (29.3–38.3)	27.7 (20.7–37.8)	0.118	
POD-1 Valine	108.4 (99.9–180.3)	126.7 (93.7–151.3)	0.797	114 (94.2–149.9)	128 (93.8–153.7)	0.287	167.6 (132.8–203.8)
POD-7 Valine	122.7 (103.7–139.9)	121.5 (105.4–146.9)	0.932	129 (107.8–150.6)	119.2 (95–138)	0.173	
POD-1 Histidine	46.7 (37.4–56.1)	47.1 (45–58.8)	0.400	49.2 (42.9–58.1)	46.6 (42.9–53.5)	0.245	51.3 (44–67.3)
POD-7 Histidine	49.4 (39–63.6)	45.2 (40.7–60.3)	0.745	46.2 (40.8–61.6)	45.2 (39.2–61.2)	0.782	
RDW-CV [%CV]	13.9 (13.3–14.75)	12.8 (12.5–13.6)	0.016 *	13.4 (12.6–14.5)	13 (12.5–14.3)	0.716	
RDW-CV >14.5	4 (33.3%)	2 (8.7%)	0.151	3 (16.7%)	3 (17.6%)	1	
RDW-SD [fL]	43.4 (40.6–48)	41.6 (38.5–45)	0.211	43.3 (40.3–45.6)	41.6 (38. 5–44.9)	0.373	
Cerebral vasospasm	10 (83.3%)	8 (34.8%)	0.011 *	18 (100%)	NA		
CVS (day)	5 (4–6)	5 (4.25–6)	1	5 (4–6)	NA		
12-month GOS	2 (1–2)	4 (4–5)	<0.001 *	2.5 (1.75–4)	4 (4–5)	<0.001 *	

CVS, cerebral vasospasm; GCS, Glasgow coma scale; GOS, Glasgow outcome scale; LOS, length of stay in hospital; POD, postoperative day; WFNS, World Federation of Neurosurgical Societies.

**Table 4 jcm-11-00380-t004:** Differential concentration of amino acids in comparison without and with Benjamini-Hochberg correction of *p*-values. Concentrations were compared using U Mann-Whitney test.

Amino Acid	POD 1 vs. Controls			POD 1 vs. POD 7			POD 1 CVS vs. No CVS			POD 1 GOS 4–5 vs. GOS 1–3			POD 7 CVS vs. No CVS			POD 7 GOS 4–5 vs. GOS 1–3		
	FC	*p*-Value	Adjusted *p*-Value	Median Difference [nmol/mL]	*p*-Value	Adjusted *p*-Value	FC	*p*-Value	Adjusted *p*-Value	FC	*p*-Value	Adjusted *p*-Value	FC	*p*-Value	Adjusted *p*-Value	FC	*p*-Value	Adjusted *p*-Value
Alanine	0.7016	<0.0001	0.0001	−15.085	0.158359	0.77752	1.0049	0.2834	0.5563	1.2351	0.357	0.9446	1.086	0.6321	0.8044	0.732	0.1305	0.9445
Arginine	0.9326	0.7868	0.8489	2.06	0.827161	0.899814	1.2021	0.2413	0.5563	0.9723	0.8213	0.9446	1.259	0.0407	0.2739	0.8293	0.1491	0.9445
Asparagine	0.9613	0.7058	0.8382	1.955	0.403521	0.77752	1.0711	0.5974	0.8731	0.9545	0.4654	0.9446	1.0433	0.7042	0.8044	0.9333	0.414	0.9861
Glutamine	1.1165	0.0932	0.1667	142.95	0.394466	0.77752	1.0538	0.3221	0.5563	1.0508	0.664	0.9446	1.0099	0.7538	0.8044	0.9717	0.7809	0.9861
Glutamic acid	0.6656	<0.0001	0.0001	5.41	0.576427	0.84247	1.2119	0.7917	0.9553	1.1815	0.4979	0.9446	1.2825	0.1057	0.37	0.8592	0.5663	0.9861
Glycine	0.6447	<0.0001	<0.0001	23.055	0.294711	0.77752	1.2596	0.1248	0.4741	0.871	0.9446	0.9446	1.2997	0.0577	0.2739	0.8403	0.3569	0.9861
Isoleucine	0.9199	0.558	0.7067	3.825	0.334075	0.77752	1.1715	0.4283	0.6781	1.1799	0.4549	0.9446	1.1392	0.5634	0.8044	1.0318	0.4444	0.9861
Leucine	0.8353	0.0154	0.0372	−9.54	0.852456	0.899814	1.1543	0.9474	1	0.9827	0.917	0.9446	1.1902	0.3639	0.6748	0.9325	0.8348	0.9861
Lysine	1.0252	0.8969	0.8969	9.28	0.480881	0.77752	1.1243	0.1131	0.4741	0.9266	0.9032	0.9446	1.0851	0.8044	0.8044	0.9001	0.7676	0.9861
Methionine	0.8441	0.0157	0.0372	−0.17333	0.279851	0.77752	1.2607	0.9605	1	0.8998	0.862	0.9446	1.4288	0.0346	0.2739	0.705	0.0734	0.9445
Phenylalanine	0.8684	0.0965	0.1667	2.195	0.265495	0.77752	1.2684	0.0515	0.4741	1.0539	0.8212	0.9446	1.2032	0.1812	0.4303	0.8549	0.4979	0.9861
Proline	0.7968	0.0073	0.0232	12.075	0.231797	0.77752	1.0548	0.6559	0.8901	1.1348	0.0681	0.9446	1.0574	0.7165	0.8044	0.8834	0.8894	0.9861
Hydroxyproline	0.614	0.0222	0.0468	−0.2	0.491065	0.77752	0.6068	0.0407	0.4741	1.1181	0.4342	0.9446	0.5021	0.0513	0.2739	1.4491	0.4237	0.9861
Serine	0.9392	0.8042	0.8489	4.786667	0.045642	0.77752	1.1596	0.1813	0.5219	0.9604	0.8757	0.9446	1.1772	0.3301	0.6748	0.9703	0.9861	0.9861
Threonine	0.9521	0.4919	0.6676	−0.36667	0.451178	0.77752	1.1531	0.1923	0.5219	0.9899	0.4549	0.9446	1.1285	0.3907	0.6748	0.9516	0.8894	0.9861
Tryptophan	0.8354	0.1951	0.2851	−3.195	0.743223	0.882577	1.0028	0.2983	0.5563	1.0289	0.1697	0.9446	1.037	0.7664	0.8044	0.9835	0.9861	0.9861
Tyrosine	0.7386	0.0001	0.0003	2.625	0.668035	0.846178	1.2137	0.1208	0.4741	1.0028	0.781	0.9446	1.2141	0.1168	0.37	0.9171	0.424	0.9861
Valine	0.7515	0.0005	0.0019	4.07	0.656279	0.846178	1.1628	1	1	0.987	0.7944	0.9446	1.224	0.1707	0.4303	0.9169	0.9446	0.9861
Histidine	0.9195	0.134	0.2122	−1.19	0.967753	0.967753	1.0721	0.8045	0.9553	1.0368	0.3945	0.9446	1.0926	0.7917	0.8044	0.8725	0.7412	0.9861

CVS, cerebral vasospasm; FC, fold change; POD, postoperative day.

**Table 5 jcm-11-00380-t005:** Performance of logistic regression models.

	Amino Acids Only (Pro-OH, Phe)			Hunt&Hess Scale Only (HH)			Combined Pro-OH, Phe, and HH		
Predictions (Odds Ratios)	Pro-OH	OR 0.76 (95%CI: 0.49–1.18)		HH	OR 6.84 (95%CI: 1.78–26.37)		Pro-OH	OR 0.91 (95%CI: 0.55–1.51)	
	Phe	OR 1.05 (95%CI: 0.99–1.11)					Phe	OR 1.11 (95%CI: 1.00–1.23)	
							HH	OR 15.47 (95%CI: 2.30–103.95)	
Confusion matrix	Prediction	Reference:		Prediction	Reference:		Prediction	Reference:	
		No CVS	CVS		No CVS	CVS		No CVS	CVS
	No CVS	15	6	No CVS	12	4	No CVS	14	1
	CVS	2	12	CVS	5	14	CVS	3	17
Accuracy	77.14% (95%CI: 59.86–89.56%)			74.29% (95%CI: 56.74–87.51%)			88.57% (95%CI: 73.26–96.80%)		
Cutoff	≥0.5455			≥0.7529			≥0.4283		
Sensitivity	66.67%			77.78%			94.44%		
Specificity	88.24%			70.59%			82.35%		
PPV	85.71%			73.68%			85.00%		
NPV	71.43%			75.00%			93.33%		
AUC ROC	0.7451 (95%CI: 0.5619–0.9283)			0.7745 (95%CI: 0.6322–0.9168)			0.9052 (95%CI: 0.8056–1.0000)		
LOOCV estimate of prediction error	0.2390			0.2015			0.1797		

AUC, area under the curve; CVS, cerebral vasospasm; HH, Hunt-Hess scale; LOOCV, leave-one-out cross-validation; NPV, negative predictive value; positive predictive value; Phe, phenylalanine; Pro-OH, hydroxyproline.

## Data Availability

The data presented in this study are available on request from the corresponding author. The data are not publicly available due to ethical restrictions.

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
