# Peer review of "Plasma Amino Acids May Improve Prediction Accuracy of Cerebral Vasospasm after Aneurysmal Subarachnoid Haemorrhage"

_jcm, 2022, doi:10.3390/jcm11020380_

Round 1

Reviewer 1 Report

Authors have conducted a prospective study on 35 patients after acute subarachnoid hemorrhage (aSAH) and 37 healthy volunteers in order to verify whether plasma concentration of amino acids have
prognostic value in predicting cerebrovascular vasospasm (CVS).  Fasting peripheral blood samples were collected on postoperative day 1 and 7. High performance liquid chromatography-mass spectrometry (HPLC-MS) analysis was performed. 18 patients from the study group developed CVS. Hydroxyproline and phenylalanine presented significant CVS prediction potential and in combination with Hunt-Hess Scale provided 
the model with the best predictive performance.

Clear limitation of this study is low number of patients, however on the positive side there is a control group which allows for certain conclusions. There are only several reports in the literature on this subject and prediction of CVS remains an important benchmark in the treatment.

The authors have assessed the plasma amino-acids on the 1 and 7 postoperative day; however not all patients get a surgery on the first day following aSAH; some of the patients experience warning leaks or present 2-3 days following onset of symptoms. One further issue is that from 35 patients, 34 were operated and only 1 patient underwent coiling. Some neurosurgical expert opinions are that an open surgery reduces the risk of CVS (since yet not proven hypothesis is that CVS arises from blood degradation products, and in open surgery blood is usually more efficiently evacuated than in coiling); so I suggest to exclude this patient from the assesment and make a comparison between operated patients and healthy volunteers and to re-write the title "Plasma amino acids MAY IMPROVE prediction accuracy of CVS after aSAH in patients who underwent surgical clipping with Hunt and Hess Grade I-III".

Introduction should be expanded on the latest information regarding prevention and treatment of CVS. I suggest to include latest articles, such as:

Kole MJ, Wessell AP, Ugiliweneza B, Cannarsa GJ, Fortuny E, Stokum JA, Shea P, Chryssikos T, Khattar NK, Crabill GA, Schreibman DL, Badjatia N, Gandhi D, Aldrich EF, James RF, Simard JM. Low-Dose Intravenous Heparin Infusion After Aneurysmal Subarachnoid Hemorrhage is Associated With Decreased Risk of Delayed Neurological Deficit and Cerebral Infarction. Neurosurgery. 2021 Feb 16;88(3):523-530. doi: 10.1093/neuros/nyaa473. PMID: 33269390.

Rumalla K, Lin M, Ding L, Gaddis M, Giannotta SL, Attenello FJ, Mack WJ. Risk Factors for Cerebral Vasospasm in Aneurysmal Subarachnoid Hemorrhage: A Population-Based Study of 8346 Patients. World Neurosurg. 2021 Jan;145:e233-e241. doi: 10.1016/j.wneu.2020.10.008. Epub 2020 Oct 10. PMID: 33049382.

Harris L, Hill CS, Elliot M, Fitzpatrick T, Ghosh A, Vindlacheruvu R. Comparison between outcomes of endovascular and surgical treatments of ruptured anterior communicating artery aneurysms. Br J Neurosurg. 2021 Jun;35(3):313-318. doi: 10.1080/02688697.2020.1812517. Epub 2020 Aug 27. PMID: 32852231.

Lines 60-62 I would leave out - in most of the centers patients with Grade IV and Grade V HH underwent surgery or endovascular intervention, regardless of their clinical state at the presentation. 

Please include the current treatment modality of CVS in the introduction with proper citation. Furthermore, pathophysiology of speculated changes in concentration of the amino acids in relation to aSAH on the molecular, pathophysiological level, should be more precisely explained (the hypermetabolic state or catabolism leading to increase of plasma amino-acid levels). 

Materials and methods are properly written and statistics explained. In the Results section the authors mention Tables, but in my version of the manuscript there are no tables - please add these. For all patients who underwent aSAH all details of the treatment should be included (which aneurysm, which surgical approach, complete/incomplete clipping, existence of further aneurysms, presence of any kind of metabolical diseases or major traumas which could lead to increase/decrease of plasma amino-acids and therefore a bias, pre- and postoperative neurological state, further complications or surgeries during the first 7 days etc). It should be clearly stated how many patients remained on the respirator for the observed period of time (i.e., 1-7 days), since prolonged imobilization and respiratory support could effect levels of plasma amino acids significantly and provide false conclusions. These levels were reported not to have been changed, but these data should however be revealed. Why were the day 1 and 7 taken as cuttofs?

Patients with aSAH featured distinctive profile of amino acids’ concentration - please explain in detail what does this mean and provide a figure compared to healthy individuals. Combination of the serum levels of aminoacids with HH scale is problematic - as this scale is the golden standard for initial assesment of the patient, its predictive value is well known so that it is unclear which additional value do the measurments of amino acids provide. What is the ration of these measurments, which positive consequences for treatment of CVS can arrise from this and very important - how expensive are these tests and is there a cost effectiveness compared to standard daily TCD measurments followed by standard CVS treatment if applicable?

Level of hydroxiproline was found to be lower in patients with aSAH and even lower in patients with CVS; what are your speculations for this finding? Is it possible that low hydroxiproline could be a predictor for possible aSAH in patients from a certain risk group (for example polycytic ovary diseases or genetic predilection)?

In line 301 authors state that their results match the results of the previous studies - however, previous studies in this paragraph are referred to measurments of amino-acids in CSF and not in plasma. Please correct and explain. Why was the measurment of amino-acids in CSF not performed in patients with aSAH and compared to plasma-levels. This would be also very interesting. Please include in the table how many patients with aSAH had an external ventricular drain, how many developed hydrocephalus and how many developed symptomatic strokes due to CVS. 

Please include following studies in the discussion and comment on each study with comparison to your findings:

1) 10.3389/fneur.2017.00438 ; Sokol et al. Amino Acids in Cerebrospinal Fluid of Patients with Aneurysmal Subarachnoid Haemorrhage: An Observational Study

2) Dhandapani S, Goudihalli S, Mukherjee KK, Singh H, Srinivasan A, Danish M, Mahalingam S, Dhandapani M, Gupta SK, Khandelwal N, Mathuriya SN. Prospective study of the correlation between admission plasma homocysteine levels and neurological outcome following subarachnoid hemorrhage: a case for the reverse epidemiology paradox? Acta Neurochir (Wien). 2015 Mar;157(3):399-407. doi: 10.1007/s00701-014-2297-0. Epub 2014 Dec 17. PMID: 25510646.

3)Kumar M, Goudihalli S, Mukherjee K, Dhandapani S, Sandhir R. Methylenetetrahydrofolate reductase C677T variant and hyperhomocysteinemia in subarachnoid hemorrhage patients from India. Metab Brain Dis. 2018 Oct;33(5):1617-1624. doi: 10.1007/s11011-018-0268-5. Epub 2018 Jun 21. PMID: 29926428.

4) Appel D, Seeberger M, Schwedhelm E, Czorlich P, Goetz AE, Böger RH, Hannemann J. Asymmetric and Symmetric Dimethylarginines are Markers of Delayed Cerebral Ischemia and Neurological Outcome in Patients with Subarachnoid Hemorrhage. Neurocrit Care. 2018 Aug;29(1):84-93. doi: 10.1007/s12028-018-0520-1. PMID: 29560598.

5) Staalsø JM, Bergström A, Edsen T, Weikop P, Romner B, Olsen NV. Low plasma arginine:asymmetric dimethyl arginine ratios predict mortality after intracranial aneurysm rupture. Stroke. 2013 May;44(5):1273-81. doi: 10.1161/STROKEAHA.111.000605. Epub 2013 Mar 5. PMID: 23463757.

Author Response

Please see the attached manuscript with tables

Response to Reviewer 1 Comments

Authors have conducted a prospective study on 35 patients after acute subarachnoid hemorrhage (aSAH) and 37 healthy volunteers in order to verify whether plasma concentration of amino acids have prognostic value in predicting cerebrovascular vasospasm (CVS).  Fasting peripheral blood samples were collected on postoperative day 1 and 7. High performance liquid chromatography-mass spectrometry (HPLC-MS) analysis was performed. 18 patients from the study group developed CVS. Hydroxyproline and phenylalanine presented significant CVS prediction potential and in combination with Hunt-Hess Scale provided the model with the best predictive performance.

Clear limitation of this study is low number of patients, however on the positive side there is a control group which allows for certain conclusions. There are only several reports in the literature on this subject and prediction of CVS remains an important benchmark in the treatment.

The authors have assessed the plasma amino-acids on the 1 and 7 postoperative day; however not all patients get a surgery on the first day following aSAH; some of the patients experience warning leaks or present 2-3 days following onset of symptoms.

All aneurysms were successfully secured within 48 hours after initial bleed. We added appropriate information in the methods.

One further issue is that from 35 patients, 34 were operated and only 1 patient underwent coiling. Some neurosurgical expert opinions are that an open surgery reduces the risk of CVS (since yet not proven hypothesis is that CVS arises from blood degradation products, and in open surgery blood is usually more efficiently evacuated than in coiling); so I suggest to exclude this patient from the assesment and make a comparison between operated patients and healthy volunteers and to re-write the title "Plasma amino acids MAY IMPROVE prediction accuracy of CVS after aSAH in patients who underwent surgical clipping with Hunt and Hess Grade I-III".

Thank you for this comment. On the other hand, the patient who underwent coiling, did not suffer from cerebral vasospasm. Recalculating the statistics now would increase the risk of type one error by unnecessarily increasing the number of false positives. While we appreciate the biological risks associated with various types of invasive procedures, in our opinion, the suggested change of the original database can increase error because of the low sample size. Interestingly, authors of the recent meta-analysis observed no difference in the risk of CVS for coiling vs. clipping procedures [Rumalla et al., 2020]. If the reviewer considers removing the patient from the database and recalculating statistics a sine qua non requirement, then we need 2 weeks for this revision, however, we believe that our explanation is sufficient.

Introduction should be expanded on the latest information regarding prevention and treatment of CVS. I suggest to include latest articles, such as:

Rumalla K, Lin M, Ding L, Gaddis M, Giannotta SL, Attenello FJ, Mack WJ. Risk Factors for Cerebral Vasospasm in Aneurysmal Subarachnoid Hemorrhage: A Population-Based Study of 8346 Patients. World Neurosurg. 2021 Jan;145:e233-e241. doi: 10.1016/j.wneu.2020.10.008. Epub 2020 Oct 10. PMID: 33049382.

Kole MJ, Wessell AP, Ugiliweneza B, Cannarsa GJ, Fortuny E, Stokum JA, Shea P, Chryssikos T, Khattar NK, Crabill GA, Schreibman DL, Badjatia N, Gandhi D, Aldrich EF, James RF, Simard JM. Low-Dose Intravenous Heparin Infusion After Aneurysmal Subarachnoid Hemorrhage is Associated With Decreased Risk of Delayed Neurological Deficit and Cerebral Infarction. Neurosurgery. 2021 Feb 16;88(3):523-530. doi: 10.1093/neuros/nyaa473. PMID: 33269390.

Harris L, Hill CS, Elliot M, Fitzpatrick T, Ghosh A, Vindlacheruvu R. Comparison between outcomes of endovascular and surgical treatments of ruptured anterior communicating artery aneurysms. Br J Neurosurg. 2021 Jun;35(3):313-318. doi: 10.1080/02688697.2020.1812517. Epub 2020 Aug 27. PMID: 32852231.

We added appropriate information and citations in the introduction.

Lines 60-62 I would leave out - in most of the centers patients with Grade IV and Grade V HH underwent surgery or endovascular intervention, regardless of their clinical state at the presentation.

We removed the indicated sentence and modified the paragraph accordingly.

Please include the current treatment modality of CVS in the introduction with proper citation.

We added appropriate information and citation [Connolly et al., Stroke, 2012] in the introduction.

Furthermore, pathophysiology of speculated changes in concentration of the amino acids in relation to aSAH on the molecular, pathophysiological level, should be more precisely explained (the hypermetabolic state or catabolism leading to increase of plasma amino-acid levels).

We added appropriate paragraph in the discussion.

Materials and methods are properly written and statistics explained. In the Results section the authors mention Tables, but in my version of the manuscript there are no tables - please add these. For all patients who underwent aSAH all details of the treatment should be included (which aneurysm, which surgical approach, complete/incomplete clipping, existence of further aneurysms, presence of any kind of metabolical diseases or major traumas which could lead to increase/decrease of plasma amino-acids and therefore a bias, pre- and postoperative neurological state, further complications or surgeries during the first 7 days etc). It should be clearly stated how many patients remained on the respirator for the observed period of time (i.e., 1-7 days), since prolonged imobilization and respiratory support could effect levels of plasma amino acids significantly and provide false conclusions. These levels were reported not to have been changed, but these data should however be revealed.

The details of the treatment are included in the Tables, which were successfully enclosed to the submission. Patients with serious systemic disease, malnutrition, altered state of consciousness, presence of malignancy, multiple aneurysms, and/or prolonged respiratory support were excluded from the study to avoid additional bias. The appropriate information is available in the methods.

Why were the day 1 and 7 taken as cuttofs?

We intended to find a change in the amino acid concentration in the blood before clinically overt CVS, hence the test at the very beginning and then at the point at which CVS is immediately expected. The appropriate information is available in the discussion.

Patients with aSAH featured distinctive profile of amino acids’ concentration - please explain in detail what does this mean and provide a figure compared to healthy individuals.

We changed the word “profile” into “concentration” to make it more clear. The detailed comparison of amino acids’ concentration in aSAH patients and healthy individuals is visualized on Figure 1A and 1B, Figure 2A, and Table 4 (the left column).

Combination of the serum levels of amino acids with HH scale is problematic - as this scale is the golden standard for initial assessment of the patient, its predictive value is well known so that it is unclear which additional value do the measurements of amino acids provide.

The superiority of combined model as provided in table 5 is sufficient to show the difference of Hunt-Hess scale alone performance and combined model performance. The addition of specific amino acids concentration to the model increased the prognostic value.

What is the ration of these measurments, which positive consequences for treatment of CVS can arrise from this and very important - how expensive are these tests and is there a cost effectiveness compared to standard daily TCD measurments followed by standard CVS treatment if applicable?

Early identification of patients at risk for CVS who could benefit from aggressive intervention is extremely important to improve neurological outcomes. The proposed measurements are not part of the basic tests performed in a clinical hospital, so the results may provide guidance for subsequent clinical studies but cannot yet be implemented in clinical practice. We added appropriate information in the discussion.

Level of hydroxiproline was found to be lower in patients with aSAH and even lower in patients with CVS; what are your speculations for this finding? Is it possible that low hydroxiproline could be a predictor for possible aSAH in patients from a certain risk group (for example polycytic ovary diseases or genetic predilection)?

Hydroxyproline deficiency can reduce mechanical strength of arterial walls and thus predispose to aneurysm formation and an increased vulnerability to vasospasm. It is possible that in patients with a genetic predisposition, blood levels of hydroxyproline may be a predictor of aSAH. We added appropriate information in the discussion.

In line 301 authors state that their results match the results of the previous studies - however, previous studies in this paragraph are referred to measurments of amino-acids in CSF and not in plasma. Please correct and explain. Why was the measurment of amino-acids in CSF not performed in patients with aSAH and compared to plasma-levels. This would be also very interesting.

We clarified in the text that we also observed decreased levels of amino acids in plasma. Unfortunately in the studied group not all the patients had CSF examination, thus we decided not to include these data to the analysis.

Please include in the table how many patients with aSAH had an external ventricular drain, how many developed hydrocephalus and how many developed symptomatic strokes due to CVS.

We excluded patients with hydrocephalus as a possible source of bias. The appropriate information was added to the methods.

Please include following studies in the discussion and comment on each study with comparison to your findings:

1) 10.3389/fneur.2017.00438 ; Sokol et al. Amino Acids in Cerebrospinal Fluid of Patients with Aneurysmal Subarachnoid Haemorrhage: An Observational Study

2) Dhandapani S, Goudihalli S, Mukherjee KK, Singh H, Srinivasan A, Danish M, Mahalingam S, Dhandapani M, Gupta SK, Khandelwal N, Mathuriya SN. Prospective study of the correlation between admission plasma homocysteine levels and neurological outcome following subarachnoid hemorrhage: a case for the reverse epidemiology paradox? Acta Neurochir (Wien). 2015 Mar;157(3):399-407. doi: 10.1007/s00701-014-2297-0. Epub 2014 Dec 17. PMID: 25510646.

3)Kumar M, Goudihalli S, Mukherjee K, Dhandapani S, Sandhir R. Methylenetetrahydrofolate reductase C677T variant and hyperhomocysteinemia in subarachnoid hemorrhage patients from India. Metab Brain Dis. 2018 Oct;33(5):1617-1624. doi: 10.1007/s11011-018-0268-5. Epub 2018 Jun 21. PMID: 29926428.

4) Appel D, Seeberger M, Schwedhelm E, Czorlich P, Goetz AE, Böger RH, Hannemann J. Asymmetric and Symmetric Dimethylarginines are Markers of Delayed Cerebral Ischemia and Neurological Outcome in Patients with Subarachnoid Hemorrhage. Neurocrit Care. 2018 Aug;29(1):84-93. doi: 10.1007/s12028-018-0520-1. PMID: 29560598.

5) Staalsø JM, Bergström A, Edsen T, Weikop P, Romner B, Olsen NV. Low plasma arginine:asymmetric dimethyl arginine ratios predict mortality after intracranial aneurysm rupture. Stroke. 2013 May;44(5):1273-81. doi: 10.1161/STROKEAHA.111.000605. Epub 2013 Mar 5. PMID: 23463757.

These are all relevant studies, thus we cited them in the discussion.

Reviewer 2 Report

The authors studied the profiles of plasma amino acids in patients after aSAH and combined with Hunt-Hess Scale to improve the diagnosis and prognosis of CVS. Overall, the experiments were well designed, and analysis were comprehensively performed. I have a few concerns listed below, which authors should address before publication:

1. The figure resolution was too low, and it is hard to clearly read the label information in the panels.
2. There is no need to illustrate all the tested amino acids in Figure 5, it looks too busy but without useful information. Showing the significant predictive potential amino acid (Phe and Pro-OH) is good enough.
3. There are too many tables in this manuscript, and some table contains similar information as showing in the figure, such as Table 5 and Figure 5. The author should concisely illustrate the useful information.
4. As only 35 patient samples were analyzed, it would be helpful if the author could use published external data to validate their findings.
5. The p-value should be provided in the figures for all the ROC analysis or any other significance analysis (such as Figure 1 B, D and Figure 4B, D).

Author Response

Please see the attached manuscript with tables.

Response to Reviewer 2 Comments

The authors studied the profiles of plasma amino acids in patients after aSAH and combined with Hunt-Hess Scale to improve the diagnosis and prognosis of CVS. Overall, the experiments were well designed, and analysis were comprehensively performed. I have a few concerns listed below, which authors should address before publication:

  1. The figure resolution was too low, and it is hard to clearly read the label information in the panels.

We have improved the figure resolution so that it is easy to read.

  1. There is no need to illustrate all the tested amino acids in Figure 5, it looks too busy but without useful information. Showing the significant predictive potential amino acid (Phe and Pro-OH) is good enough.

We have reduced the content of Figure 5 and given the corresponding p-values.

  1. There are too many tables in this manuscript, and some table contains similar information as showing in the figure, such as Table 5 and Figure 5. The author should concisely illustrate the useful information.

We have removed Table 5 and given the corresponding p-values on Figure 5.

  1. As only 35 patient samples were analyzed, it would be helpful if the author could use published external data to validate their findings.

So far, our search has revealed that there is no public repository with similar data that could be used for external validation. We plan to externally validate our results in the next independent patient cohort, although we believe it is crucial to show current results now.

  1. The p-value should be provided in the figures for all the ROC analysis or any other significance analysis (such as Figure 1 B, D and Figure 4B, D).

We have added information to the figure legend that the relevant p-values are available in Table 4.

Round 2

Reviewer 1 Report

The authors have sufficiently answered the remarks of the reviewer. 

Reviewer 2 Report

The author addressed my concerns!